# MetaShift: A Dataset of Datasets for Evaluating Contextual Distribution Shifts and Training Conflicts

**Weixin Liang**
Stanford University
`wxliang@stanford.edu`

**James Zou**
Stanford University
`jamesz@stanford.edu`

## Abstract

Understanding the performance of machine learning models across diverse data distributions is critically important for reliable applications. Motivated by this, there is a growing focus on curating benchmark datasets that capture distribution shifts. While valuable, the existing benchmarks are limited in that many of them only contain a small number of shifts and they lack systematic annotation about what is different across different shifts. We present MetaShift—a collection of 12,868 sets of natural images across 410 classes—to address this challenge. We leverage the natural heterogeneity of Visual Genome and its annotations to construct MetaShift. The key construction idea is to cluster images using its metadata, which provides context for each image (e.g. *cats with cars* or *cats in bathroom*) that represent distinct data distributions. MetaShift has two important benefits: first, it contains orders of magnitude more natural data shifts than previously available. Second, it provides explicit explanations of what is unique about each of its data sets and a distance score that measures the amount of distribution shift between any two of its data sets. We demonstrate the utility of MetaShift in benchmarking several recent proposals for training models to be robust to data shifts. We find that the simple empirical risk minimization performs the best when shifts are moderate and no method had a systematic advantage for large shifts. We also show how MetaShift can help to visualize conflicts between data subsets during model training [1].

## 1 Introduction

A major challenge in machine learning (ML) is that a model can have very different performances and behaviors when it's applied to different types of natural data (Koh et al., 2020; Izzo et al., 2021; 2022). For example, if the user data have different contexts compared to the model's training data (e.g. users have outdoor dog photos and the model's training was mostly on indoor images), then the model's accuracy can greatly suffer (Yao et al., 2022). A model can have disparate performances even within different subsets within its training and evaluation data (Daneshjou et al., 2021; Eyuboglu et al., 2022). In order to assess the reliability and fairness of a model, we therefore need to evaluate its performance and training behavior across heterogeneous types of data. However, the lack of well-structured datasets representing diverse data distributions makes systematic evaluation difficult.

In this paper, we present MetaShift to tackle this challenge. MetaShift is a collection of 12,868 sets of natural images from 410 classes. Each set corresponds to images in a similar context and represents a coherent real-world data distribution, as shown in Figure 1. The construction of MetaShift is different from and complementary to other efforts to curate benchmarks for data shifts by pulling together data across different experiments or sources. MetaShift leverages heterogeneity within the large sets of images from the Visual Genome project (Krishna et al., 2017) by clustering the images using metadata that describes the context of each image. The advantage of this approach is that MetaShift contains many more coherent sets of data compared to other benchmarks. Importantly, we have explicit annotations of what makes each subset unique (e.g. *cats with cars* or *dogs next to a bench*) as

---

[1]Dataset and code available at: `https://metashift.readthedocs.io/`

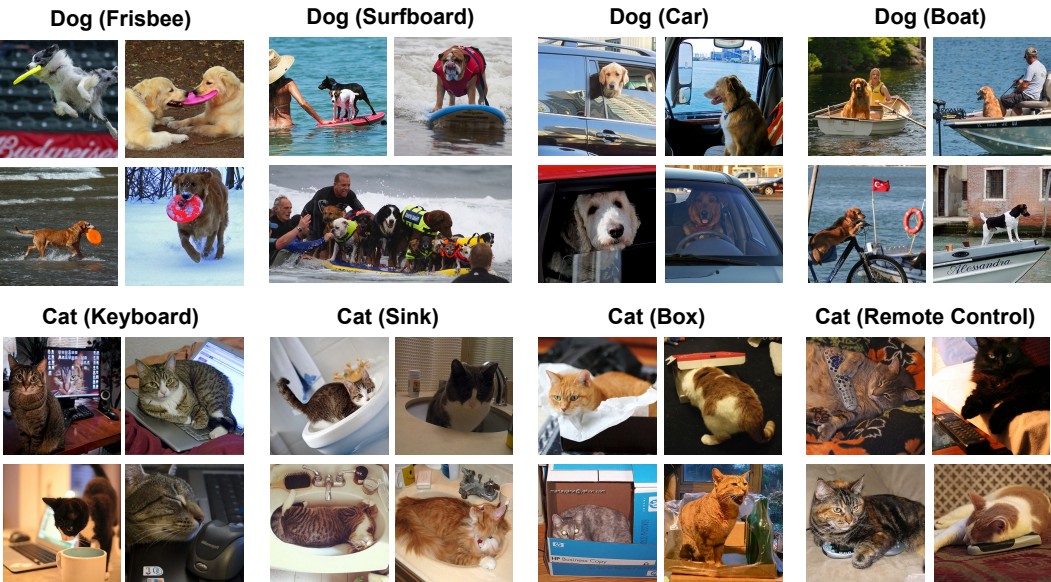

Figure 1: **Example subsets of natural images from MetaShift.** MetaShift leverages the natural heterogeneity within each class (e.g., "cat", "dog") to provide many subsets of images. Each subset corresponds to images in a similar context (the context is stated in parenthesis) and represents a coherent real-world data distribution. Here, we only show 2 out of 410 classes and 8 out of 12,868 subsets of images from MetaShift.

well as a score that measures the distance between any two subsets, which is not available in previous benchmarks of natural data.

We demonstrate the utility of MetaShift in two applications. First, MetaShift supports evaluation on both *domain generalization* and *subpopulation shifts* settings. Using the score between subsets provided by MetaShift, we study ML models' behavior under different carefully modulated amounts of distribution shift. Second, MetaShift can also shed light on the training dynamics of ML models. Since we have the subset membership information for each training datum, we could attribute the contribution of each gradient step back to the training subsets, and then analyze how different data subsets provide conflicting training signals.

**Our contributions:** We present MetaShift as an important resource for studying the behavior of ML algorithms and training dynamics across data with heterogeneous contexts. Our methodology for constructing MetaShift can also be applied to other domains where metadata is available. We empirically evaluate the performance of different robust learning algorithms, showing that ERM performs well for modest shifts while no method is the clear winner for larger shifts. This finding suggests that domain generalization is an important and challenging task and that there's still a lot of room for new methods.

## 2 RELATED WORK

**Existing Benchmarks for Distribution Shift**    Distribution shifts have been a longstanding challenge in machine learning. Early benchmarks focus on distribution shifts induced by synthetic pixel transformations. Examples include rotated and translated versions of MNIST and CIFAR (Worrall et al., 2017); surface variations such as texture, color, and corruptions like blur in Colored MNIST (Gulrajani & Lopez-Paz, 2020), ImageNet-C (Hendrycks & Dietterich, 2019). Although the synthetic pixel transformations are well-defined, they generally do not represent realistic shifts in real-world images that we capture in MetaShift.

Other benchmarks do not rely on transformations but instead pull together data across different experiments or sources. Office-31 (Saenko et al., 2010) and Office-home (Venkateswara et al., 2017) contain images collected from different domains like Amazon, clipart. These benchmarks typically have only a handful of data distributions. The benchmarks collected in WILDS (Koh et al., 2020) combine data from different sources (e.g., medical images from different hospitals, animal images

from different camera traps). Similarly, some meta-learning benchmarks (Triantafillou et al., 2019; Guo et al., 2020) focuses on dataset-level shift by combining different existing datasets like ImageNet, Omniglot. While valuable, they lack systematic annotation about what is different across different shifts. Santurkar et al. (2020); Ren et al. (2018) utilize the hierarchical structure of ImageNet to construct training and test sets with disjoint subclasses. For example, the "tableware" class uses "beer glass" and "plate" for training and testing respectively. Different from their work, we study the shifts where the core object remains the same while the context changes. NICO (He et al., 2020) query different manually-curated phrases on search engines to collect images of objects in different contexts. A key difference is the scale of MetaShift: NICO contains 190 sets of images across 19 classes while MetaShift has 12,868 sets of natural images across 410 classes.

To sum up, the advantages of our MetaShift are:

- Existing benchmark datasets for distribution shifts typically have only a handful of data distributions. In contrast, our MetaShift has over 12,868 data distributions, thus enabling a much more comprehensive assessment of distribution shifts.
- Distribution shifts in existing benchmarks are not annotated (i.e. we don't know what drives the shift) and are not well-controlled (i.e. we can't easily adjust the magnitude of the shift). The MetaShift provides explicit annotations of the differences between any two sub-datasets, and it quantifies the distance of the shift.

**Evaluating conflicts on training data**   Recent work has shown that different training data points play a heterogeneous role in training. To quantify this, Data Shapley (Ghorbani & Zou, 2019; Kwon & Zou, 2022) provides a mathematical framework for quantifying the contribution of each training datum. Data Cartography (Swayamdipta et al., 2020) leverages a model's training confidence to discover hard-to-learn training data points. Such understanding has provided actionable insights that benefit the ML workflow. For example, removing noisy low-contribution training data points improves the model's final performance (Liang et al., 2020a; 2021). Furthermore, active learning identifies the most informative data points for humans to annotate (Liang et al., 2020b). Complementary to their work, our analysis sheds light on not only which but also why a certain portion of the training data are hard-to-learn—because different subsets are providing *conflicting* training signals.

## 3   THE METASHIFT CONSTRUCTION METHODOLOGY

What is MetaShift? The MetaShift is a collection of subsets of data together with an annotation graph that explains the similarity/distance between two subsets (edge weight) as well as what is unique about each subset (node metadata). For each class, say "cat", we have many subsets of cats, and we can think of each subset as a node in the graph, as shown in Figure 2. Each subset corresponds to "cat" in a different context: e.g. "cat with sink" or "cat with fence". The context of each subset is the node metadata. The "cat with sink" subset is more similar to "cat with faucet" subset because there are many images that contain both sink and faucet. This similarity is the weight of the edge; a higher weight means the contexts of the two nodes tend to co-occur in the same data.

How can we use MetaShift? It is a flexible framework to generate a large number of real-world distribution shifts that are well-annotated and controlled. For each class of interest, say "cats", we can use the meta-graph of cats to identify a collection of cats nodes for training (e.g. cats with bathroom-related contexts) and a collection of cats nodes for out-of-domain evaluation (e.g. cats in outdoor contexts). Our meta-graph tells us exactly what is different between the train and test domains (e.g. bathroom vs. outdoor contexts), and it also specifies the similarity between the two contexts via graph distance. That makes it easy to carefully modulate the amount of distribution shift. For example, if we use cats-in-living-room as the test set, then this is a smaller distribution shift.

**Base Dataset: Visual Genome**   We leverage the natural heterogeneity of Visual Genome and its annotations to construct MetaShift. Visual Genome contains over 100k images across 1,702 object classes. For each image, Visual Genome annotates the class labels of all objects that occur in the image. Formally, for each image $x^{(i)}$, we have a list of meta-data tags $m^{(i)} = \{t_1^{(i)}, t_2^{(i)}, \ldots, t_{n_m}^{(i)}\}$, each indicating the presence of an object in the context. We denote the vocabulary of the meta-data tags as $\mathbb{M} = \{m_0, \ldots, m_{|\mathbb{M}|}\}$. MetaShift is constructed on a class-by-class basis: For each class, say "cat", we pull out all cat images and proceed with the following steps.

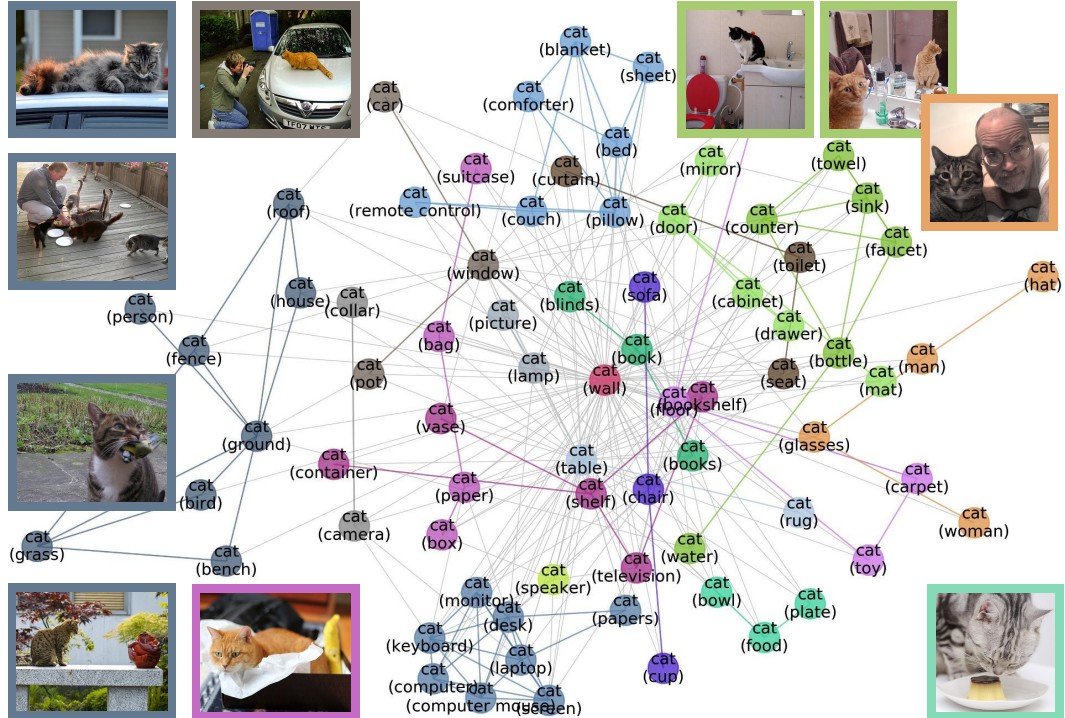

Figure 2: **Meta-graph—visualizing the diverse data distributions within the "cat" class.** Each node represents one subset of the cat images. Each subset corresponds to "cat" in a different context: e.g. "cat with sink" or "cat with fence". Each edge indicates the similarity between the two connecting subsets. Node colors indicate the communities automatically detected by graph-based algorithms. Inter-community edges are colored and intra-community edges are grayed out for better visualization. The border color of each example image indicates its community in the meta-graph. We have one such meta-graph for each of the 410 classes in the MetaShift. Beyond visualization, meta-graph also provides a natural and systematic way to quantify the distance between any two subsets (i.e., nodes), which is not available in previous benchmarks of natural data.

**Step 1: Generate Candidate Subsets** We first generate candidate subsets by enumerating all possible meta-data tags. We construct $|\mathbb{M}|$ candidate subsets where the $i^{th}$ subset contains all images of the class of interest (i.e., "cat") that has a meta-tag $m_i$. We then remove subsets whose sizes are less than a threshold (e.g., 25).

**Step 2: Construct Meta-graphs** Since the meta-data are not necessarily disentangled, the candidate subsets might contain significant overlaps (e.g., "cat with sink" and "cat with faucet"). To capture this phenomenon, we construct a meta-graph to model the relationships among all subsets of each class. Specifically, for each class $j \in \mathbb{Y}$, we construct meta-graph, a weighted undirected graph $\mathcal{G} = (\mathcal{V}, \mathcal{E})$ where each node $v \in \mathcal{V}$ denotes a candidate subset, and the weight of each edge is the overlap coefficient between two subsets:

$$\text{overlap}(X, Y) = \frac{|X \cap Y|}{\min(|X|, |Y|)}, \tag{1}$$

We remove the edges whose weights are less than a threshold (e.g., 0.1) to sparsify the graph. As shown in Figure 2, the meta-graph $\mathbb{G}$ captures meaningful semantics of the multi-modal data distribution of the class of interest.

**Step 3: Quantify Distances of Distribution Shifts** The geometry of meta-graphs provides a natural and systematic way to quantify the distances of shifts across different data distributions: Intuitively, if two subsets are far away from each other in the MetaGraph, then the shift between them tend to be large. Following this intuition, we leverage *spectral embeddings* (Belkin & Niyogi, 2003; Chung & Graham, 1997) to assign an embedding for each node based on the graph geometry. Spectral

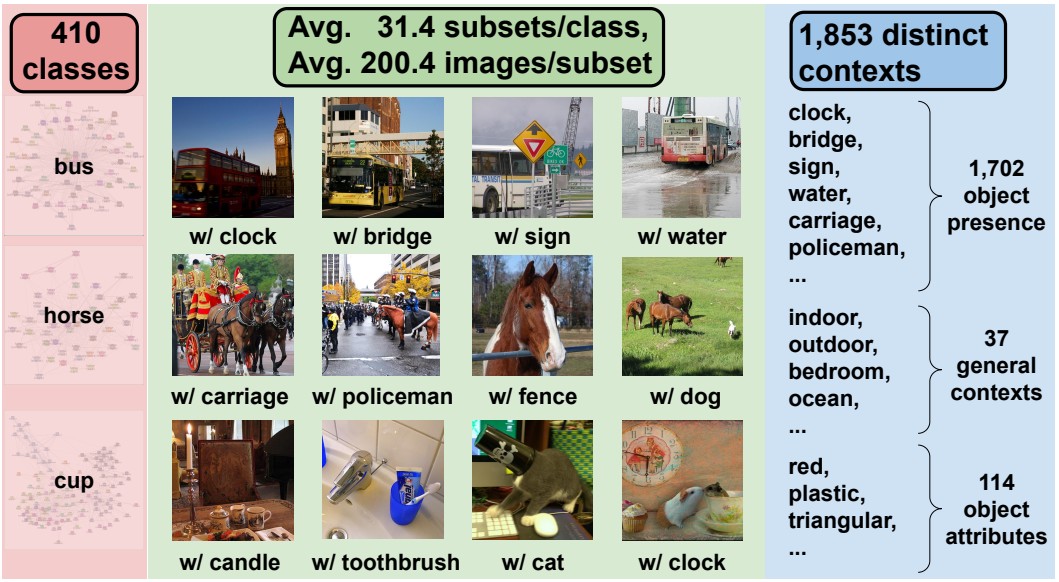

Figure 3: **Summary of MetaShift.** MetaShift covers a wide range of 410 classes and 12,868 sets of natural images in total. For each class, we have 31.4 subsets on average together with an annotation graph (i.e., meta-graph) that explains the similarity/distance between two subsets (edge weight) as well as what is unique about each subset (node metadata). More concretely, the subsets are characterized by a diverse collection of 1,853 distinct contexts, which covers 1,702 object presence, 37 general contexts and 114 object attributes.

embedding minimizes the expected square distance between nodes that are connected:

$$\min_{X: X^T 1 = 0, X^T X = I_K} \sum_{i,j \in V} A_{ij} \|X_i - X_j\|^2 \tag{2}$$

where $X_i$ is the embedding for node $i \in \mathcal{V}$ and $K$ is the dimension of the embedding, and A is the adjacency matrix. We denote by $X$ the matrix of dimension $n \times K$ whose $i$-th row $X_i$ corresponds to the embedding of node $i$. The constraint $X^T 1 = 0$ forces the embedding to be centered and $X^T X = I_K$ ensures that we do not get trivial solution like all node embeddings located at the origin (i.e., $X = 0$). Denoting by $L = D - A$ the Laplacian matrix of the graph, we have:

$$\text{tr}(X^T L X) = \frac{1}{2} \sum_{i,j \in V} A_{ij} \|X_i - X_j\|^2 \tag{3}$$

$$\min_{X: X^T 1 = 0, X^T X = I_K} \text{tr}(X^T L X) = \sum_{k=2}^{K+1} \lambda_k \tag{4}$$

The minimum is reached for $X$ equal to the matrix of eigenvectors of the Laplacian matrix associated with the eigenvalues $\lambda_2, ..., \lambda_{K+1}$. After calculating the spectral embeddings, we use the euclidean distance between the embeddings of two nodes as their distance. Other off-the-shelf node embedding methods like Node2Vec (Grover & Leskovec, 2016) can be readily plugged into MetaShift. We delay exploring them as future work.

Although the subset overlap (i.e., edge weight) can also be used as a similarity metric, it does not incorporate the structural information from neighboring nodes. Our node embedding-based distance captures not only such overlap, but also broader similarities.

**Step 4: Simulating Distribution Shifts** MetaShifts allows users to benchmark both (1) domain generalization and (2) subpopulation shifts in a well-annotated (explicit annotation of what drives the shift) and well-controlled (easy control of the amount of distribution shift) fashion.

- In *domain generalization*, the train and test distributions comprise data from related but distinct domains. This arises in many real-world scenarios since it is often infeasible to construct a comprehensive training set that spans all domains. To simulate this setting, we can sample two

distinct collections of subsets as the train domains and the test domains respectively (e.g. bathroom vs. outdoor contexts). To adjust the magnitude of the shift, we can fix the test domains and change the train domains with different distances. For example, if we use cats-in-living-room as the test set, then this is a smaller distribution shift.

- In *subpopulation shifts*, the train and test distributions are mixtures of the same domains, but the mixture weights change between train and test. It is also an important setting since ML models are often reported to perform poorly on under-represented groups. To simulate this setting, we can sample the training set and test set from the same subsets but with different mixture weights. To adjust the magnitude of the shift, we can use different mixture weights for the training set while keeping the test set unchanged.

We demonstrate both settings in the experiment section. It is also worth noting that different subsets may share common images—e.g. a dog image can have both grass and frisbee would occur in both *dog with grass* and *dog with frisbee*. Therefore, a post-processing step is needed to remove the training images that also occur in the test set to ensure no data leakage.

**MetaShift Statistics**  Figure 3 shows the statistics of MetaShift across all tasks. We start from the pre-processed and cleaned version of Visual Genome (Hudson & Manning, 2019), which contains 113,018 distinct images across 1,702 object classes. After the dataset construction, we have 12,868 sets of natural images from 410 classes. Concretely, each class has 31.4 subsets, and each subset has 200.4 images on average.

The subsets are characterized by a diverse vocabulary of 1,853 distinct contexts. Beyond 1,702 contexts defined by object presence, MetaShift also leverages the 37 distinct general contexts and 114 object attributes from Visual Genome. The general contexts typically describe locations (i.e., indoor, outdoor), weather (e.g., rainy, cloudless), and places (e.g., bathroom, ocean). The object attributes include color (e.g., white, red), material (e.g., wood, plastic), shape (e.g., round, square), and other object-specific properties (e.g., empty/full for plates). See Appendix A for examples and more information.

**Generalizability: Case Study on COCO**  The MetaShift construction methodology is quite simple, and can be extended to any dataset with metadata tags (i.e., multilabel). To demonstrate this, we apply the construction methodology on MS-COCO dataset (Lin et al., 2014), which provides object detection labels for each image. Applying our construction methodology, we are able to construct 1321 subsets, with an average size of 389 and a median size of 124, along with 80 meta-graphs that help to quantify the amount of distribution shift among MS-COCO subsets. Experiments on MetaShift from COCO can be found in Appendix D.

Even though the data that we used here (e.g. Visual Genome, COCO) is not new, the idea of turning one dataset into a structured collection of datasets for assessing distribution shift is less explored. COCO (Lin et al., 2014) and many other datasets provide meta-data that could be naturally used for generating candidate subsets. The *main contribution* of our methodology is that after generating the candidate subsets, our method provides a meta-graph that could be used to determine the train and test domains, and also specifies the similarity between the two subsets via graph distance. MetaShift opens up a new window for systematically evaluating domain shifts.

## 4 EXPERIMENT

We demonstrate the utility of MetaShift in two applications:

- **Evaluating distribution shifts:** MetaShift supports evaluating both *domain generalization* (Section 4.1) and *subpopulation shifts* (Section 4.2). We perform controlled studies on ML models' behavior under different amounts of distribution shift. We also evaluate several recent proposals for training models to be robust to data shifts.
- **Assessing training conflicts:** The subset information in MetaShift also sheds light on the training dynamics of ML models (Section 4.3). Since we have the subset membership information for each training datum, we could attribute the contribution of each gradient step back to the training subsets, and then analyze the heterogeneity of the contributions made by different training subsets.

| Algorithm | Task 1: Cat vs. Dog | | | | Task 2: Bus vs. Truck | | | | Task 3: Elephant vs. Horse | | | | Task 4: Bowl vs. Cup | | | |
|---|---|---|---|---|---|---|---|---|---|---|---|---|---|---|---|---|
| | $d$=0.44 | $d$=0.71 | $d$=1.12 | $d$=1.43 | $d$=0.81 | $d$=1.20 | $d$=1.42 | $d$=1.52 | $d$=0.44 | $d$=0.63 | $d$=0.89 | $d$=1.44 | $d$=0.16 | $d$=0.47 | $d$=1.03 | $d$=1.31 |
| ERM | **0.844** | 0.605 | 0.357 | 0.240 | **0.950** | 0.863 | 0.702 | 0.609 | **0.964** | 0.821 | 0.793 | 0.729 | **0.888** | 0.768 | 0.401 | 0.276 |
| IRM | 0.814 | **0.628** | 0.380 | **0.341** | 0.863 | **0.901** | 0.752 | 0.634 | 0.943 | 0.886 | 0.764 | 0.750 | 0.883 | **0.793** | 0.426 | **0.404** |
| GroupDRO | 0.837 | 0.597 | 0.434 | 0.264 | 0.901 | 0.857 | **0.770** | **0.665** | 0.936 | 0.864 | 0.829 | 0.743 | 0.829 | 0.765 | 0.444 | 0.303 |
| CORAL | 0.798 | 0.589 | **0.481** | 0.302 | 0.925 | 0.801 | 0.783 | 0.640 | 0.929 | **0.900** | 0.814 | 0.771 | 0.850 | 0.734 | **0.482** | 0.274 |
| CDANN | 0.729 | 0.620 | 0.380 | 0.326 | 0.944 | 0.888 | 0.789 | 0.584 | 0.921 | 0.857 | **0.836** | **0.779** | 0.879 | 0.752 | 0.432 | 0.280 |

Table 1: **Evaluating domain generalization:** out-of-domain test accuracy with different amounts of distribution shift $d$. Higher $d$ indicates more challenging problem. Test data are fixed and only training data are changed.

## 4.1 EVALUATING DOMAIN GENERALIZATION

In *domain generalization*, the train and test distributions comprise data from related but distinct domains. To benchmark domain generalization using MetaShift, we can sample two distinct collections of subsets as the train domains and the test domains respectively. We also showcase adjusting the magnitude of the shift by sampling different training subsets while keeping the test set unchanged.

**Setup** As shown in figure 2, subsets like "cat with sink" and "cat with faucet" are quite similar to each other. To make our evaluation settings more challenging, we merge similar subsets by running Louvain community detection algorithm (Blondel et al., 2008) on each meta-graph. Node color in Figure 2 indicates the community detection result. The embeddings of the merged subset are calculated as the average embeddings of subsets being merged, weighted by the subset size. After merging similar subsets, we construct four binary classification tasks:

1. **Cat vs. Dog:** Test on dog(*shelf*) with 129 images. The cat training data is cat(*sofa + bed*) (i.e., cat(*sofa*) unions cat(*bed*)). We keep the test set and the cat training data unchanged. We keep the total size of training data as 400 images unchanged. We explore the effect of using 4 different sets of dog training data:
    (a) dog(*cabinet + bed*) as dog training data. Its distance to dog(*shelf*) is $d$=0.44
    (b) dog(*bag + box*) as dog training data. Its distance to dog(*shelf*) is $d$=0.71
    (c) dog(*bench + bike*) as dog training data. Its distance to dog(*shelf*) is $d$=1.12
    (d) dog(*boat + surfboard*) as dog training data. Its distance to dog(*shelf*) is $d$=1.43
2. **Bus vs. Truck:** Test on truck(*airplane*) with 161 images. The bus training data is bus(*clock + traffic light*). The 4 different sets of truck training data are truck(*cone + fence*), truck(*bike + mirror*), truck(*flag + tower*), truck(*traffic light + dog*). Their distances to truck(*airplane*) are $d$=0.81, $d$=1.20, $d$=1.42, $d$=1.52, respectively, as shown in Table 1.
3. **Elephant vs. Horse:** Test on horse(*barn*) with 140 images. The elephant training data is elephant(*fence + rock*). The 4 different sets of horse training data are horse(*dirt + trees*), horse(*fence + helmet*), horse(*car + wagon*), horse(*statue + cart*), with distances $d$=0.44/0.63/0.89/1.44.
4. **Bowl vs. Cup:** Test on cup(*coffee*) with 375 images. The bowl training data is bowl(*fruits + tray*). The 4 different sets of horse training data are cup(*knife + tray*), cup(*water + cabinet*), cup(*computer + lamp*), cup(*toilet + box*), with distances $d$=0.16/0.47/1.03/1.31 respectively.

**Domain generalization algorithms** Following recent benchmarking efforts (Koh et al., 2020; Gulrajani & Lopez-Paz, 2020), we evaluated several recent methods for training models to be robust to data shifts: IRM (Arjovsky et al., 2019), GroupDRO (Sagawa et al., 2020), CORAL (Sun & Saenko, 2016), CDANN (Long et al., 2018). To ensure a fair comparison, we use the implementation and the default hyperparameters from DomainBed (Gulrajani & Lopez-Paz, 2020). We discuss more experiment setup details in Appendix B.1. We also extend the binary classification experiment here to a 10-class classification experiment in Appendix B.2.1.

**Results** Our experiments show that the standard empirical risk minimization performs the best when shifts are moderate(i.e., when $d$ is small). For example, ERM consistently achieves the best test accuracy on the smallest $d$ of all tasks. This finding is aligned with previous research, as the domain generalization methods typically impose strong algorithmic regularization in various forms (Koh et al., 2020; Santurkar et al., 2020). However, when the amount of distribution shift increases, the domain generalization algorithms typically outperform the ERM baseline, though no algorithm

| Algorithm | Average Acc. | | | Worst Group Acc. | | |
|---|---|---|---|---|---|---|
| | $p$=12% | $p$=6% | $p$=1% | $p$=12% | $p$=6% | $p$=1% |
| ERM | **0.840** | 0.823 | 0.759 | 0.715 | 0.701 | 0.562 |
| IRM | 0.825 | **0.830** | 0.773 | 0.729 | 0.715 | 0.528 |
| GroupDRO | 0.837 | 0.818 | 0.762 | 0.701 | **0.715** | 0.576 |
| CORAL | 0.835 | 0.816 | 0.755 | 0.667 | 0.694 | 0.597 |
| CDANN | 0.836 | 0.810 | **0.774** | **0.750** | 0.701 | **0.625** |

Table 2: **Evaluating subpopulation shift:** $p$ is the percentage of minority groups in the training data. Lower $p$ indicates a more challenging setting. Evaluated on a balanced test set ($p$=50%). The classification task is "Cat vs. Dog" with "indoor/outdoor" as the spurious correlation.

is a consistent winner compared to other algorithms for large shifts. This finding suggests that domain generalization is an important and challenging task and that there's still a lot of room for new methods.

## 4.2 Evaluating Subpopulation Shifts

In *subpopulation shifts*, the train and test distributions are mixtures of the same domains with different mixture weights. This is a more frequently-encountered problem since real-world datasets often have minority groups, while standard models are often reported to perform poorly on under-represented demographics (Buolamwini & Gebru, 2018; Koenecke et al., 2020).

**Setup** To benchmark subpopulation shifts using MetaShift, we can sample two distinct collections of subsets as the minority groups and majority groups respectively. We then use different mixture weights to construct the training set and test set. For "Cat vs. Dog", we leverage the general contexts "indoor/outdoor" which have a natural spurious correlation with the class labels. Concretely, in the training data, cat(*ourdoor*) and dog(*indoor*) subsets are the minority groups, while cat(*indoor*) and dog(*outdoor*) are majority groups. We keep the total size of training data as 1700 images unchanged and only vary the portion of minority groups. We use a balanced test set with 576 images to report both average accuracy and worst group accuracy. We extend the binary classification experiment here to a 10-class classification experiment in Appendix B.2.2.

**Results** Table 2 shows the evaluation results on subpopulation shifts. In terms of the average accuracy, ERM performs the best when $p$ is large (i.e., the minority group is less underrepresented) as expected. However, in terms of the worst group accuracy, the algorithms typically outperform ERM, showcasing their effectiveness in subpopulation shifts. Furthermore, it is worth noting that no algorithm is a consistent winner compared to other algorithms for large shifts. This finding is aligned with our finding in the previous section, suggesting that subpopulation shift is an important and challenging problem and there's still a lot of room for new methods.

## 4.3 Assessing Training Conflicts

Next, we demonstrate how MetaShift can shed light on the training dynamics of ML models. An important feature of MetaShift is that each training datum is not only associated with a class label, but also the annotations of subset membership. Such annotations open a window for a systematic evaluation of how training on each subset would affect the evaluation performance on other subsets.

**Setup** Concretely, we evaluate the ML model after *each gradient step*. We record the change of the validation loss of each validation subset. This gives us a multi-axes description of the effect of the current gradient step. Meanwhile, we also record the subset membership information for each of the training data in the current batch. We then run a linear regression to analyze the contributions of each training subset to the change of the validation loss of each validation subset.

We sampled 48 subsets from cat & dog classes. We use 22 of them as out-of-domain evaluation sets and the other 26 as training and in-domain evaluation. We sample 50 images from each subset to form the training set. Figure 4 shows the abridged results for brevity and better visualization.

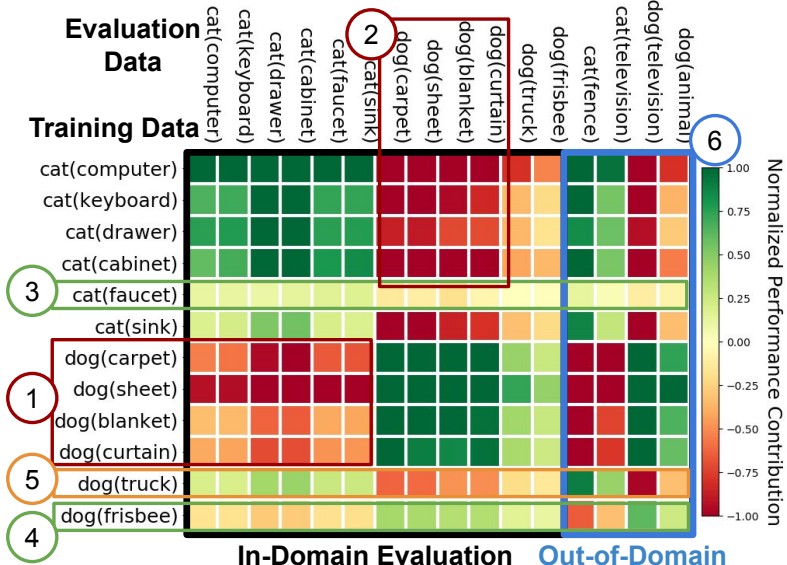

Figure 4: **A Visualization of Training Conflicts.** Each row represents a training subset, and each column represents an evaluation subset. A positive value (green) indicates that training on the training subset would improve the performance on the evaluation subset. A negative value (red) indicates that the training would hurt the performance on the evaluation subset. See text for discussions.

**Results** Region (1) and (2) in Figure 4 highlighted the "conflict zones": indoor cat subsets and indoor dog subsets are having severe training conflicts with each other. This makes sense since they share the same spurious feature "indoor", making the model confused. However, such training conflicts are important for out-of-domain generalization. As highlighted in Region (6), the dog(*animal*) subset (last column, which comprises primarily outdoor images) benefits most from indoor subsets like dog(*sheet*), rather than outdoor dog subsets like dog(*truck*). These results might indicate that training conflicts force the ML model to learn generalizable features.

Furthermore, Figure 4 also reveals the subsets that have low training contributions. As highlighted in Region (3) and (4), training on cat(*faucet*) and dog(*frisbee*) has little effect on other subsets. We hypothesize that this is because these subsets are too easy and thus provide little insight for the model. To verify this hypothesis, we remove cat(*faucet*), and dog(*frisbee*) from training and use them for out-of-domain validation. We found that the model achieves a near perfect accuracy on both subsets: cat(*faucet*): 98.1%, dog(*frisbee*): 96.6%. This shows that subsets with low training contributions tend to be the subsets that are too easy.

Other observations include: Columns of cat(*television*) and dog(*television*) in Region (6) look opposite to each other. In addition, dog(*truck*) as highlighted in (5) exhibits a very different behaviour compared to other dog subsets: training on dog(truck) conflicts with all other dog subsets.

## 5 CONCLUSION

We present MetaShift—a collection of 12,868 sets of natural images from 410 classes—as an important resource for studying the behavior or ML algorithms and training dynamics across data with heterogeneous contexts. MetaShift contains orders of magnitude more natural data shifts than previously available. It also provides explicit explanations of what is unique about each of its data sets and a distance score that measures the amount of distribution shift between any two of its data sets. We demonstrate the utility of MetaShift in both evaluating distribution shifts, and visualizing conflicts between data subsets during model training. Even though the data that we used here (e.g. Visual Genome) is not new, the idea of turning one dataset into a structured collection of datasets for assessing distribution shift is less explored. Our methodology for constructing MetaShift is simple and powerful, and can be readily extended to create many new resources for evaluating distribution shifts, which is very much needed.

## REPRODUCIBILITY STATEMENT

Our project website `https://MetaShift.readthedocs.io/` provides detailed documentation and installation guidelines. The dataset and code are also available at `https://github.com/Weixin-Liang/MetaShift`. The implementations will enable researchers to download MetaShift and reproduce the results described here as well as run their own evaluations on additional datasets. In the experiments of evaluating domain generalization and subpopulation shifts, we use the implementation and the default hyperparameters from Domainbed (Gulrajani & Lopez-Paz, 2020) to ensure a fair comparison. Our MetaShift dataset and code are available in the supplementary material, and would use the Creative Commons Attribution 4.0 International License.

## ETHICS STATEMENT

One limitation is that our MetaShift might inherit existing biases in Visual Genome, which is the base dataset of our MetaShift. Potential concerns include minority groups being under-represented in certain classes (e.g., women with snowboard), or annotation bias where people in images are by default labeled as male when gender is unlikely to be identifiable. Existing work in analyzing, quantifying, and mitigating biases in general computer vision datasets can help with addressing this potential negative societal impact. We can also use MetaShift to understand how certain subsets of training data can lead to model biases in other contexts, as done in our experiments in Figure 4.

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

# A    ADDITIONAL DATASET INFORMATION

For each image class (e.g. *Dogs*), the MetaShift dataset contains different sets of dogs under different contexts to represent diverse data distributions. The contexts include presence/absence of other objects (e.g. *dog with frisbee*). Contexts can also reflect attributes (e.g. *black dogs*) and general settings (e.g. *dogs in sunny weather*). These concepts thus capture diverse and real-world distribution shifts. We list the attribute and general location contexts below.

## A.1    GENERAL LOCATION AND ATTRIBUTE CONTEXTS

### A.1.1    GENERAL LOCATION CONTEXTS

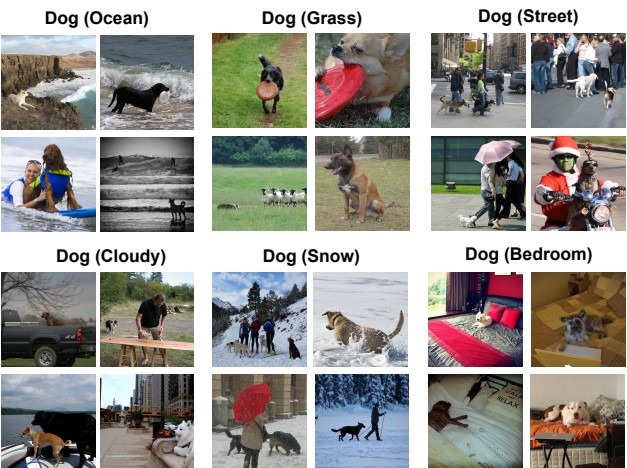

Figure 5: **Example subsets based on general contexts** (the global context is stated in parenthesis). MetaShift covers global contexts including location (e.g., indoor, outdoor) and weather (e.g., sunny, rainy).

```
GENERAL_CONTEXT_ONTOLOGY = {
    'indoor/outdoor': ['indoors', 'outdoors'],
    'weather': ['clear', 'overcast', 'cloudless', 'cloudy', 'sunny', 'foggy', 'rainy'],
    'room': ['bedroom', 'kitchen', 'bathroom', 'living room'],
    'place': ['road', 'sidewalk', 'field', 'beach', 'park', 'grass',
              'farm', 'ocean', 'pavement',
              'lake', 'street', 'train station', 'hotel room',
              'church', 'restaurant', 'forest', 'path',
              'display', 'store', 'river', 'yard',
              'snow', 'airport', 'parking lot']
}
```

Figure 6: **The general contexts and their ontology in MetaShift.** MetaShift covers 37 general contexts including location (e.g., indoor, outdoor, ocean, snow) and weather (e.g., couldy, sunny, rainy).

### A.1.2 ATTRIBUTE CONTEXTS

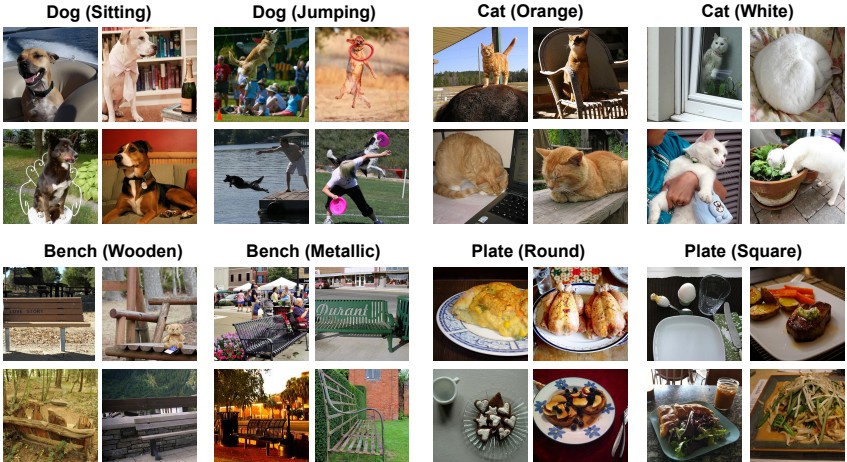

Figure 7: **Example Subsets based on object attribute contexts** (the attribute is stated in parenthesis). MetaShift covers attributes including activity (e.g., sitting, jumping), color (e.g., orange, white), material (e.g., wooden, metallic), shape (e.g., round, square), and so on.

```
ATTRIBUTE_CONTEXT_ONTOLOGY = {
 'darkness': ['dark', 'bright'], 'dryness': ['wet', 'dry'],
 'colorful': ['colorful', 'shiny'], 'leaf': ['leafy', 'bare'],
 'emotion': ['happy', 'calm'], 'sports': ['baseball', 'tennis'],
 'flatness': ['flat', 'curved'], 'lightness': ['light', 'heavy'],
 'gender': ['male', 'female'], 'width': ['wide', 'narrow'],
 'depth': ['deep', 'shallow'], 'hardness': ['hard', 'soft'],
 'cleanliness': ['clean', 'dirty'], 'switch': ['on', 'off'],
 'thickness': ['thin', 'thick'], 'openness': ['open', 'closed'],
 'height': ['tall', 'short'], 'length': ['long', 'short'],
 'fullness': ['full', 'empty'], 'age': ['young', 'old'],
 'size': ['large', 'small'], 'pattern': ['checkered', 'striped', 'dress', 'dotted'],
 'shape': ['round', 'rectangular', 'triangular', 'square'],
 'activity': ['waiting', 'staring', 'drinking', 'playing', 'eating', 'cooking', 'resting',
              'sleeping', 'posing', 'talking', 'looking down', 'looking up', 'driving',
              'reading', 'brushing teeth', 'flying', 'surfing', 'skiing', 'hanging'],
 'pose': ['walking', 'standing', 'lying', 'sitting', 'running', 'jumping', 'crouching',
          'bending', 'smiling', 'grazing'],
 'material': ['wood', 'plastic', 'metal', 'glass', 'leather', 'leather', 'porcelain',
              'concrete', 'paper', 'stone', 'brick'],
 'color': ['white', 'red', 'black', 'green', 'silver', 'gold', 'khaki', 'gray',
           'dark', 'pink', 'dark blue', 'dark brown',
           'blue', 'yellow', 'tan', 'brown', 'orange', 'purple', 'beige', 'blond',
           'brunette', 'maroon', 'light blue', 'light brown']
}
```

Figure 8: **The attributes and their ontology in MetaShift.** MetaShift covers over 100 attributes including activity (e.g., sitting, jumping), color (e.g., orange, white), material (e.g., wooden, metallic), shape (e.g., round, square) and so on.

### A.2 COMPUTE REQUIREMENTS

All experiments were performed on an Amazon EC2 P3.8 instance with 4 NVIDIA V100 Tensor Core GPUs. Each training trial in evaluating distribution shifts takes less than an hour to finish.

### A.3 DATASET LICENSES

Our MetaShift and the code would use the Creative Commons Attribution 4.0 International License. Visual Genome (Krishna et al., 2017) is licensed under a Creative Commons Attribution 4.0 International License. MS-COCO (Lin et al., 2014) is licensed under CC-BY 4.0. The Visual Genome dataset uses 108, 077 images from the intersection of the YFCC100M (Thomee et al., 2016) and MS-COCO. We use the pre-processed and cleaned version of Visual Genome by GQA (Hudson & Manning, 2019).

We use the implementation and hyper-parameters from Domainbed (Gulrajani & Lopez-Paz, 2020) for empirical risk minimization (ERM) and all domain generalization baselines. Domainbed is licensed under an MIT License. It is worth noting that the implementation of empirical risk minimization (ERM) in Domainbed implicitly up-samples minority domains during training. Nevertheless, we stick with their implementation for consistency.

## B  Experiment Details

### B.1  Experiment Setup Details

In the experiments of evaluating domain generalization and subpopulation shifts, we use the implementation and the default hyperparameters from **DomainBed** (Gulrajani & Lopez-Paz, 2020) to ensure a fair comparison. More specifically, we evaluated the following methods for training models to be robust to data shifts:

- **ERM:** the standard Empirical risk minimization baseline.
- **IRM:** Invariant risk minimization (Arjovsky et al., 2019), which penalizes feature distributions that have different optimal linear classifiers for each domain.
- **GroupDRO:** Group Distributionally robust optimization (Sagawa et al., 2020), which uses distributionally robust optimization to explicitly minimize the loss on the worst-case domain during training.
- **CORAL:** Correlation Alignment (Sun & Saenko, 2016), which penalizes differences in the means and covariances of the feature distributions for each domain.
- **CDANN:** Conditional Adversarial Domain Adaptation (Long et al., 2018), which uses adversarial learning to penalize differences in feature representations from different domains.

Following DomainBed (Gulrajani & Lopez-Paz, 2020), we use ResNet-18 with the Adam optimizer, cross-entropy loss, learning rate 0.00005, batch size 32. The data augmentation steps include the following: crops of random size and aspect ratio, resizing to $224 \times 224$ pixels, random horizontal flips, random color jitter, grayscaling the image with 10% probability, and normalization using the ImageNet channel means and standard deviations. We start from ImageNet pre-trained weights, which is consistent with the standard practice of prior work (Gulrajani & Lopez-Paz, 2020; Koh et al., 2020; Sagawa et al., 2020). For additional implementation details, we refer readers to DomainBed (Gulrajani & Lopez-Paz, 2020).

### B.2  Multi-class Classification Experiments

To demonstrate the generalizability of our findings, we extend our binary classification experiments into a multi-class classification setting. More specifically, we augment the "Cat vs. Dog" classification task into a 10-class animal classification task by incorporating the following classes: "bear", "bird", "cow", "elephant", "horse", "sheep", "giraffe", "zebra".

### B.2.1  Multi-class Classification: Domain Generalization

| Algorithm | OOD Acc. | | | |
|---|---|---|---|---|
| | $d$=0.44 | $d$=0.71 | $d$=1.12 | $d$=1.43 |
| ERM | **0.762** | 0.637 | 0.448 | 0.265 |
| IRM | 0.699 | 0.660 | 0.422 | 0.284 |
| GroupDRO | 0.690 | **0.683** | **0.451** | 0.252 |
| CORAL | 0.641 | 0.624 | 0.373 | 0.261 |
| CDANN | 0.676 | 0.647 | 0.392 | **0.304** |

Table 3: **Evaluating domain generalization on a 10-class animal classification task**. Out-of-domain (OOD) test accuracy with different amounts of distribution shift $d$. Higher $d$ indicates more challenging problem.

We extend Task 1 in the domain generalization experiment (Section 4.1) into a multi-class classification setting. Overall, these new results are very consistent with our findings. The multi-class

classification setting is constructed as follows: we extend the "Cat vs. Dog" task to a 10-class animal classification task. Similar to Section 4.1, we only change the training data of dog, while keeping the training data of all other classes unchanged, in order to avoid introducing additional confounding effects.

We explore the effect of using 4 different sets of dog training data. The test is performed on dog(*shelf*), and thus we also report the distance between the dog training data and the test data dog(*shelf*).

1. dog(*cabinet + bed*) as dog training data. Its distance to dog(*shelf*) is $d$=0.44
2. dog(*bag + box*) as dog training data. Its distance to dog(*shelf*) is $d$=0.71
3. dog(*bench + bike*) as dog training data. Its distance to dog(*shelf*) is $d$=1.12
4. dog(*boat + surfboard*) as dog training data. Its distance to dog(*shelf*) is $d$=1.43

**Results** As shown in Table 3, in the multi-class classification setting, the standard empirical risk minimization still performs the best when shifts are moderate(i.e., when $d$ is small), which is aligned with previous research, as the domain generalization methods typically impose strong algorithmic regularization in various forms (Koh et al., 2020; Santurkar et al., 2020). However, when the amount of distribution shift increases, the domain generalization algorithms typically outperform the ERM baseline, though no algorithm is a consistent winner compared to other algorithms for large shifts. This finding suggests that domain generalization is an important and challenging task and that there's still a lot of room for new methods.

### B.2.2 MULTI-CLASS CLASSIFICATION: SUBPOPULATION SHIFT

We extend the "Cat vs. Dog" subpopulation Shift to a 10-class animal classification task. Similarly, to avoid introducing additional confounding effects, we only vary the training data of cat and dog in different experiments, but keep the training data of other classes unchanged across experiments. We evaluate only on the cat and dog classes. The rationale of evaluating these two classes (instead of the other classes) is that we vary the training data of these classes, and thus these classes would be the most directly impacted.

| Algorithm | Average Acc. | | | Worst Group Acc. | | |
|---|---|---|---|---|---|---|
| | $p$=12% | $p$=6% | $p$=1% | $p$=12% | $p$=6% | $p$=1% |
| ERM | **0.789** | 0.769 | 0.703 | 0.646 | 0.638 | 0.339 |
| IRM | 0.780 | 0.764 | **0.728** | 0.614 | 0.543 | 0.443 |
| GroupDRO | 0.785 | **0.778** | 0.726 | 0.630 | **0.669** | 0.520 |
| CORAL | 0.787 | 0.764 | 0.713 | 0.614 | 0.543 | 0.386 |
| CDANN | 0.783 | 0.773 | 0.709 | **0.685** | 0.591 | **0.567** |

Table 4: **Evaluating subpopulation shift in a 10-class animal classification task:** $p$ is the percentage of minority groups in the training data. Lower $p$ indicates a more challenging setting. Evaluated on a balanced test set ($p$=50%).

**Results** Table 4 shows the evaluation results on subpopulation shifts in the multi-class classification setting. In terms of the average accuracy, ERM performs the best when $p$ is large (i.e., the minority group is less underrepresented) as expected. However, in terms of the worst group accuracy, the algorithms typically outperform ERM, showcasing their effectiveness in subpopulation shifts. Furthermore, it is worth noting that no algorithm is a consistent winner compared to other algorithms for large shifts. This finding is aligned with our finding in the previous section, suggesting that subpopulation shift is an important and challenging problem and there's still a lot of room for new methods.

### B.3 APPLICATION: ASSESSING TRAINING CONFLICTS

As data is the fuel powering artificial intelligence, it is important to understand the training conflicts on heterogeneous training data. We next demonstrate the utility of MetaDataset on visualizing how different subsets of data provide conflicting training signals. The key idea is to analyze the change of

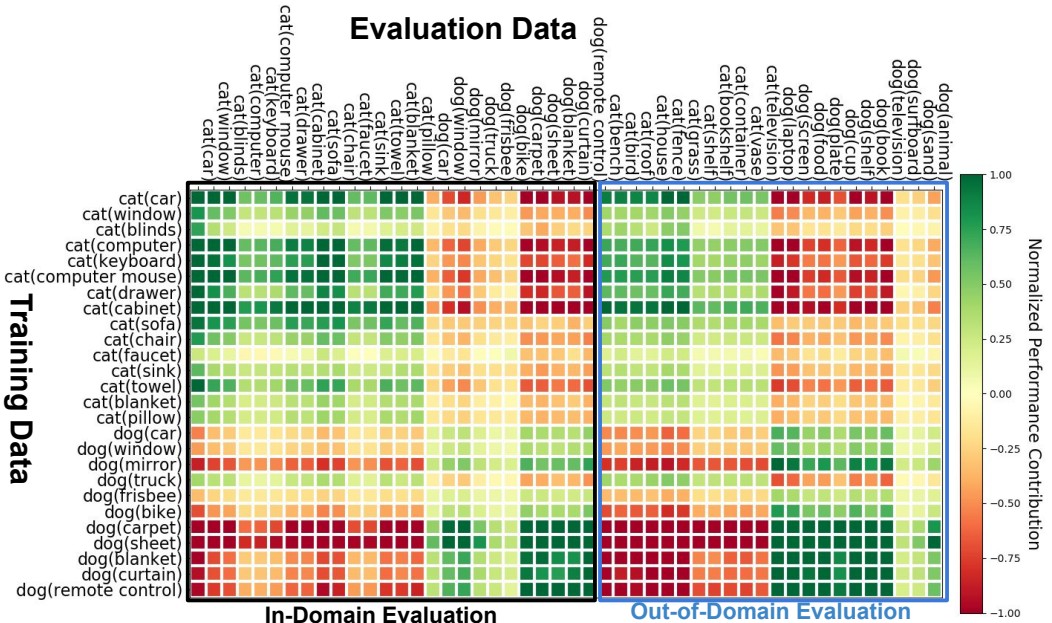

Figure 9: **The full visualization of the training conflicts experiment.** Each row represents a training subset, and each column represents an evaluation subset. A positive value (green) indicates that training on the training subset would improve the performance on the evaluation subset. A negative value (red) indicates that the training would hurt the performance on the evaluation subset. See text for discussions. Coefficients are normalized to $[-1, 1]$.

validation loss on each validation subset after each gradient step, and then attribute the change to each training subset.

As shown in the "in-domain evaluation" region of Figure 9, we randomly sample 26 cat & dog subsets for training and in-domain evaluation: we split the data in each subset into a training set and an in-domain evaluation set. As shown in the "out-of-domain evaluation" region of Figure 9, we additionally sample 22 subsets as out-of-domain evaluation sets. We downsample the training data to make sure that every training subset has an equal number of data points (i.e., 50).

## C   META-GRAPH AND META-DATA

### C.1   VISUALIZATION OF META-GRAPH SPECTRAL EMBEDDINGS

To provide a more intuitive understanding of the meta-graph embeddings, we visualize the spectral embeddings of all nodes in the "Cat" meta-graph in Figure 10. Overall, the spectral embeddings work well as similar nodes are close in the t-SNE visualization. Nodes with similar color (i.e., assigned to the same cluster by the Louvain community detection algorithm) are also clustered in the t-SNE visualization. Finally, we also note that the geometry of the t-SNE visualization (Figure 10) also closely mirrors the geometry of the meta-graph visualization layout in Figure 2.

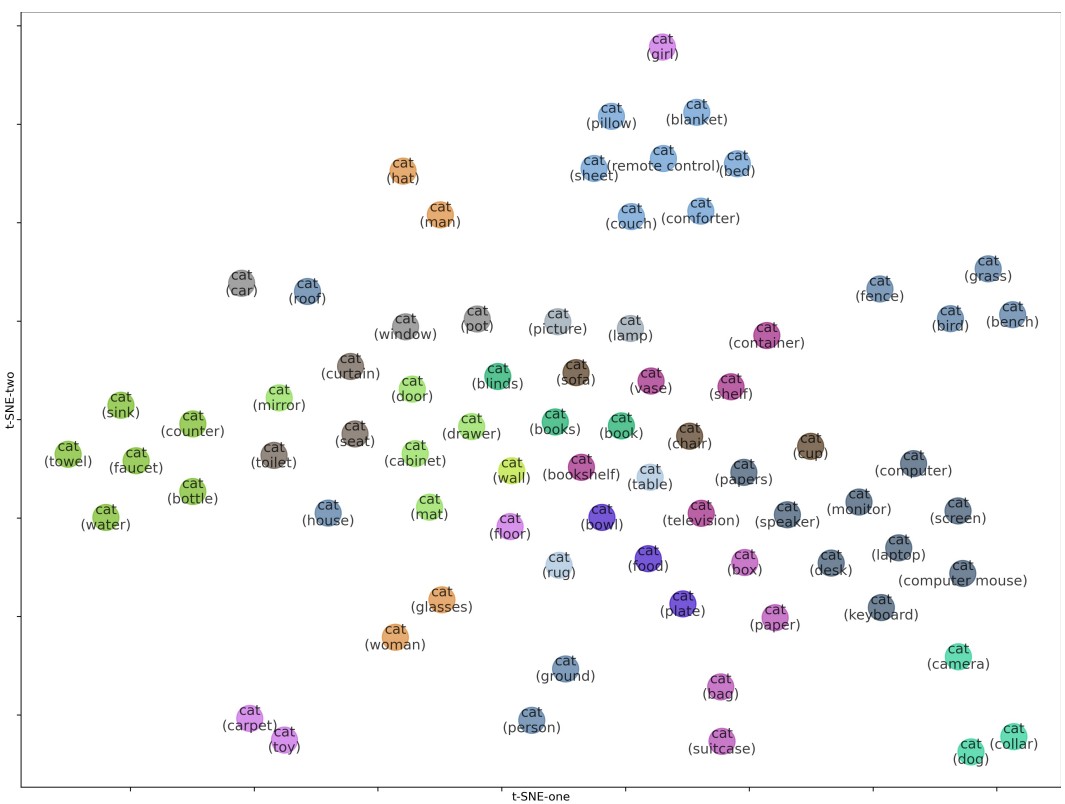

Figure 10: **t-SNE Visualization of Meta-graph Spectral Embeddings for "Cat".** Each node represents one subset of the cat images. Node colors indicate the communities automatically detected by graph-based algorithms. Overall, the spectral embeddings work well as similar nodes are close in the t-SNE visualization. Nodes with similar color (i.e., assigned to the same cluster by the Louvain community detection algorithm) are also clustered in the t-SNE visualization. Finally, we also note that the geometry of the t-SNE visualization (Figure 10) also closely mirrors the geometry of the meta-graph visualization layout in Figure 2.

### C.2   LOUVAIN COMMUNITY DETECTION ALGORITHM

Community detection algorithms detect groups of nodes in a graph that are more densely connected internally than with the rest of the graph. We apply the Louvain community detection algorithm to recover the community structure of meta-graphs. One measure of how well a network is partitioned into communities is *Modularity*, which is the difference between the actual number of edges in a community and the expected number of such edges. We apply the Louvain community detection algorithm (Blondel et al., 2008), which greedily maximizes modularity since optimizing modularity is NP-hard (Brandes et al., 2007). This is a bottom-up algorithm: initially, every vertex belongs to a separate community, and vertices are moved between communities iteratively in a way that maximizes the vertices' local contribution to the overall modularity score. When a consensus is reached (i.e. no single move would increase the modularity score), every community in the original graph is shrank to a single vertex (while keeping the total weight of the adjacent edges) and the process continues on

the next level. The algorithm stops when it is not possible to increase the modularity anymore after shrinking the communities to vertices.

### C.3 DISCUSSIONS ON IMPERFECT OR INCOMPLETE META-DATA

Even if there is some noise introduced by the imperfect or incomplete metadata, the noise can make the classification tasks more challenging and more realistic, as in subpopulation shifts. For example, it is rare that someone would collect a dog dataset of 100% indoor images, but it is quite possible that the dataset contains 80% indoor images and 20% outdoor images. Therefore, noise from metadata would not invalidate our approach, but make the curated dataset more realistic and challenging.

## D CONSTRUCTING METASHIFT FROM COCO DATASET

**MetaShift from COCO**  To demonstrate the generalizability of the MetaShift construction methodology, we apply our methodology on MS-COCO dataset (Lin et al., 2014). We construct 1321 subsets, with an average size of 389 and a median size of 124, along with 80 meta-graphs that help to quantify the amount of distribution shift among MS-COCO subsets.

**Subpopulation Shift Setup**  Based on the MetaShift from COCO, we constructed a subpopulation shift experiment similar to Section 4.2. We construct a "Cat vs. Dog" task, where the "indoor/outdoor" contexts are spuriously correlated with the class labels. The "indoor" context is constructed by merging the following super-categories: 'indoor', 'kitchen', 'electronic', 'appliance', 'furniture'. Similarly, the "outdoor" context is constructed by merging the following super-categories: 'outdoor', 'vehicle', 'sports'. In addition, in the training data, cat(*ourdoor*) and dog(*indoor*) subsets are the minority groups, while cat(*indoor*) and dog(*outdoor*) are majority groups. We keep the total size of training data as 3000 images unchanged and only vary the portion of minority groups. We use a balanced test set with 524 images to report both average accuracy and worst group accuracy.

| Algorithm | Average Acc. | | | Worst Group Acc. | | |
|---|---|---|---|---|---|---|
| | $p$=13% | $p$=7% | $p$=1% | $p$=13% | $p$=7% | $p$=1% |
| ERM | 0.826 | **0.821** | 0.739 | 0.722 | **0.698** | 0.452 |
| IRM | 0.805 | 0.794 | **0.791** | 0.765 | 0.672 | 0.458 |
| GroupDRO | 0.821 | 0.811 | 0.742 | 0.730 | 0.667 | 0.500 |
| CORAL | 0.824 | 0.801 | 0.750 | **0.809** | 0.695 | **0.565** |
| CDANN | **0.834** | 0.805 | 0.748 | 0.794 | 0.690 | 0.527 |

Table 5: **Evaluating subpopulation shift on MetaShift from COCO.** $p$ is the percentage of minority groups in the training data. Lower $p$ indicates a more challenging setting. Evaluated on a balanced test set ($p$=50%).

**Subpopulation Shift Result**  As shown in Table 5, overall, we found that the new results are consistent with our previous findings. When $p$ is large (i.e., the minority group is less underrepresented), ERM outperforms most of the other algorithms in terms of average accuracy. It is also worth noting that no algorithm is a consistent winner compared to other algorithms for large shifts. These new experiments also show that MetaShift can flexibly accommodate other source datasets including COCO.

