# OpenReview forum: "MetaShift: A Dataset of Datasets for Evaluating Contextual Distribution Shifts and Training Conflicts"
_ICLR.cc/2022/Conference — ICLR 2022 Poster_

### Official Review · Reviewer_T6b1 · 2021-11-02

**Correctness:** 4
**Technical Novelty And Significance:** 3
**Empirical Novelty And Significance:** 3
**Recommendation:** 6
**Confidence:** 3

**Main Review:**

### Pros
+ The paper is well written and easy to follow.
+ The proposed approach to leverage metadata (form previously published large-scale dataset) to create datasets of domain shifts is simple and well motivated. Splitting a large dataset (with multiple labels) in a meaningful ways to study dataset shift is not trivial. However the authors came up with an intuitive (and relatively simple) approach.
+ The problem of studying shifts in dataset distribution is very relevant and important to machine learning. This dataset can benefit the community by allowing a more system evaluation of dataset shifts.

### Cons
- It would be nice to have more descriptions of the methods used to benchmark the dataset (ERM, IRM, DRO, CORAL and CDANN), architectures used (which model? pretrained or from scratch? what is the model capacity?) and training details (which loss, which optimization, learning rate, batch size, etc). If not enough space on the main text, these informations could be added on appendix.
- It would be nice if the authors would give more detail on how the embeddings of meta-graphs are computed. For example what is the matrix A and how are the embeddings computed? Why using spectral embeddings specifically rather than other approaches (eg, using the word embedding (pretrained on large language corpus) of each context)?
- It could be nice to show some more quantitative or qualitative (eg t-SNE) for the meta-graph embeddings (used to compute the shift between datasets).
- The paper states it generates >12K datasets (across 410 classes), however experiments are done only on a very tiny number of datasets (cat/dog, bus/truck. elephant/horse and bowl/cup). The proposed dataset would be much more useful to the community if the authors would provide a much larger subset of “pre-made” datasets for easy experimentation.


**Summary Of The Paper:**

In this paper, the authors introduce a new dataset (actully, a collection of datasets) called MetaShift. MetaShift is built on top of Visual Genome and leverages its metadata to cluster images, thus providing a context for each image (labels are of the form class+context, eg,  ‘cat in grass’, ‘dog in bathroom’). This context is then used to generate dataset shifts. Besides been much larger than similar (openly available) datasets, MetaShift explicitly provides the context, which can be used to compute a “distance score” pf distribution shift between any two datasets.

**Summary Of The Review:**

I am inclined to accept this paper because of (i) the simplicity of the approach to generate the datasets and (ii) the usefulness to the community. However, I would be more confident with acceptance if the authors would address the weaknesses of the paper (se above).

---

> ### Author Response · Authors · 2021-11-16
> **Response to Reviewer T6b1**
>
> Thank you for your helpful comments!
>
> **Q: t-SNE visualization of meta-graph embeddings**
>
> **A:** Following your recommendation, we have added the t-SNE visualization of the spectral embeddings of the “Cat” meta-graph (Figure 10 in Appendix C.1). Overall, the t-SNE visualization shows that the spectral embeddings work well. We found that (1) similar nodes are close in the t-SNE visualization, (2) nodes with similar color (i.e., assigned to the same cluster by the Louvain community detection algorithm) are also clustered in the t-SNE visualization.
>
>
>
> **Q: Easy Dataset Download**
>
> **A:** The MetaShift dataset is readily available for download. As stated in the reproducibility statement, we have constructed a [project website](https://metashift.readthedocs.io/) https://MetaShift.readthedocs.io/.
> The project website provides detailed documentation and installation guidelines. In particular, the “meta-data/full-candidate-subsets.pkl” file provides the Image IDs for all subsets in MetaShift. We will put the URL of the project website below the title in our camera-ready version. We believe this will be a useful resource for the ML community.
>
>
>
> **Q: Clarification on experiment details: architecture used, hyper-parameters, algorithm details**
>
> **A:** Following your recommendation, we have added Appendix B.1 to discuss more experiment setup details. For each algorithm (e.g., IRM, GroupDRO), we have added additional descriptions. In terms of the implementation of the algorithms, we use the default implementation and the hyperparameters from DomainBed (Gulrajani & Lopez-Paz, 2020) to enable a fair comparison. Following Domainbed, we use ResNet-18 with the Adam optimizer, cross-entropy loss, learning rate 0.00005, batch size 32. The data augmentation steps include the following: crops of random size and aspect ratio, resizing to 224 × 224 pixels, random horizontal flips, random color jitter, grayscaling the image with 10% probability, and normalization using the ImageNet channel means and standard deviations. We start from ImageNet pre-trained weights, which is consistent with the standard practice of prior work (Gulrajani & Lopez-Paz, 2020; Koh et al., 2020; Sagawa et al., 2020). All these clarifications have been included in Appendix B.1.
>
>
> **Q: Clarification on the technical approaches for meta-graph embedding**
>
> **A:** We choose spectral embeddings instead of using word embeddings for the meta-graph embedding because the spectral embedding is more general, and only requires the geometry of the meta-graph as input. Therefore, even if a metadata tag has unknown semantic meaning, we can still calculate its embedding based on the geometry of the meta-graph.
>
> For the notations, matrix A is the graph adjacency matrix with edge weights. The spectral embeddings are calculated as an optimization problem: We want to assign the node embeddings such that the assignment minimizes the expected square distance between nodes that are connected (i.e., close nodes are mapped to similar embeddings), and also satisfying standard regularization constraints. Additional details can be found in [1].
>
> **Reference**
>
> [1] Belkin, Mikhail, and Partha Niyogi. "Laplacian eigenmaps for dimensionality reduction and data representation." Neural computation 15.6 (2003): 1373-1396.

---

### Official Review · Reviewer_ameR · 2021-11-03

**Correctness:** 3
**Technical Novelty And Significance:** 3
**Empirical Novelty And Significance:** 3
**Recommendation:** 6
**Confidence:** 4

**Main Review:**

**[Strengths]** Good idea to study the impact of distribution shift.
+ This paper is well written and easy to understand. Section 3 gives a good introduction to the step-by-step construction of MetaShift.
+ The main advantage of MetaShift is it contains systematic annotation about the differences between different shifts. This further helps to study the effects of distribution shifts (e.g., subpopulation shifts)
+ The idea of generating a large number of real-world distribution shifts that are well-annotated and controlled is attractive. The proposed MetaShift is well illustrated and the figures are helpful to know the information of MetaShift.
+ Section 4.3 is interesting and the results give some insights.

**[Weaknesses]** The major concern is about the experimental evaluation where the constructed tasks are only binary classification.
- In Section 4.1, this work constructs four binary classification tasks to study the impact of the shift under the generalization setting. One question is, how about constructing more challenging tasks which involve more classes?
- When evaluating subpopulation shifts, the tasks are also binary which contain spurious correlation. The same question is multi-class classification tasks might be needed.
- MetaShift is a collection of 12,868 sets of natural images from 410 classes. Why do the experiments only focus on binary classification (e.g., cat vs. dog, bus vs. truck)? In another word, it seems the same settings in Sec. 4 can be constructed based on other classes. It would be helpful if this work could discuss this. Again, more challenging multi-classification setting would be very useful.

**Summary Of The Paper:**

This work proposes a collection called MetaShift to study the impact of dataset distribution. The major advantage of MetaShift is that it provides annotation/information to measure the amount of distribution shift between any two of its data sets. In the experiment, this work constructs two applications, 1) evaluating distribution shifts, assessing training conflicts.

**Summary Of The Review:**

This work introduces a good way to study the effects of distribution shifts. Specifically, this work proposes a framework called Metashift, which contains systematic annotation about the differences between different shifts. However, the major concern is about the constructed tasks in the experiment. More explanation/discussion can be included to eliminate the question.

---

> ### Author Response · Authors · 2021-11-16
> **Multi-class experiment results added for both domain generalization and subpopulation shift**
>
> Thank you for your thoughtful suggestions and helpful feedback. Following your recommendation, we have added new experiments in multi-class classification settings, for both (1) domain generation and (2) subpopulation shift, as attached below. Overall, these new results are very consistent with our findings. We have added the new results and discussions in Appendix B.2.
>
> **Table 1: Evaluating domain generalization on a 10-class classification problem.** Out-of-domain (OOD) test accuracy with different amounts of distribution shift d.
>
> | Algorithm 	| d=0.44    	| d=0.71    	| d=1.12    	| d=1.43    	|
> |-----------	|-----------	|-----------	|-----------	|-----------	|
> | ERM       	| **0.762** 	| 0.637     	| 0.448     	| 0.265     	|
> | IRM       	| 0.699     	| 0.660     	| 0.442     	| 0.284     	|
> | GroupDRO  	| 0.690     	| **0.683** 	| **0.451** 	| 0.252     	|
> | CORAL     	| 0.641     	| 0.624     	| 0.373     	| 0.261     	|
> | CDANN     	| 0.676     	| 0.647     	| 0.392     	| **0.304** 	|
>
>
>
> **Table 2: Average accuracy of subpopulation shift in a 10-class classification problem.** p is the percentage of minority groups in the training data.
>
> | Algorithm 	| p=12%     	| p=6%      	| p=1%      	|
> |-----------	|-----------	|-----------	|-----------	|
> | ERM       	| **0.789** 	| 0.769     	| 0.703     	|
> | IRM       	| 0.780     	| 0.764     	| **0.728** 	|
> | GroupDRO  	| 0.785     	| **0.778** 	| 0.726     	|
> | CORAL     	| 0.787     	| 0.764     	| 0.713     	|
> | CDANN     	| 0.783     	| 0.773     	| 0.709     	|
>
>
> **Table 3: Worst group accuracy of subpopulation shift in a 10-class classification problem.** p is the percentage of minority groups in the training data.
>
> | Algorithm 	| p=12%     	| p=6%      	| p=1%      	|
> |-----------	|-----------	|-----------	|-----------	|
> | ERM       	| 0.646     	| 0.638     	| 0.339     	|
> | IRM       	| 0.614     	| 0.543     	| 0.443     	|
> | GroupDRO  	| 0.630     	| **0.669** 	| 0.520     	|
> | CORAL     	| 0.614     	| 0.543     	| 0.386     	|
> | CDANN     	| **0.685** 	| 0.591     	| **0.567** 	|
>
>
> **Setup**: We extended the “Cat vs. Dog” task into a 10-class animal classification problem, by adding “bear”, “bird”, “cow”, “elephant”, “horse”, “sheep”, “giraffe”, “zebra”. For domain generalization, we only vary the training data of cat and dog in different experiments, but keep the training data of other classes unchanged across experiments, in order to avoid introducing additional confounding effects. Similarly, for distribution shifts, we only vary the training data of cat and dog, while keeping other classes unchanged.
>
> **Results**: Overall, we found that the new results are consistent with our previous findings.  For domain generalization, the standard empirical risk minimization still performs the best when shifts are moderate(i.e., when $d$ is small). However, when the amount of distribution shift increases, the domain generalization algorithms typically outperform the ERM baseline, though no algorithm is a consistent winner compared to other algorithms for large shifts.
>
> Similarly, for subpopulation shifts, ERM performs the best in terms of the average accuracy, when $p$ is large (i.e., the minority group is less underrepresented) as expected. However, in terms of the worst group accuracy, the algorithms typically outperform ERM, showcasing their effectiveness in subpopulation shifts. Still, it is worth noting that no algorithm is a consistent winner compared to other algorithms for large shifts.
>
> Together, these findings suggest that domain generalization and subpopulation shift are important and challenging tasks, and that there’s still a lot of room for new methods. These new experiments also show that MetaShift can flexibly accommodate multi-class predictions.
>
> We hope these new results address your concerns and you’d consider increasing your score. Please let us know if you have any further questions. Thank you again!

---

> ### Author Response · Authors · 2021-11-22
> **Dear ameR: we'd love to hear if you have any further Qs after our response**
>
> Dear reviewer ameR
>
> Thank you very much for your helpful feedback and suggestions, they helped us to improve the paper. We tried to carefully address all of your comments in our response and the updated paper. Please let us know if you have any further questions, and we are very happy to follow up!
>
> Thank you for your time!

---

> > ### Comment · Reviewer_ameR · 2021-11-29
> > **Thanks for the response**
> >
> > Dear authors,
> >
> > Thanks for the newly introduced experiment of 10-way classification, which helps address my major concern.
> >
> > I have carefully read the revised paper and I have one question on Section B.2.1:
> > In the experiment of 10-way classification, 1) the training set contains images from 10 classes, 2) test images are from Dog class only, and 3) by varying the training data of dog, the results in Table 3 are reported.
> > Please help me understand your setting and check if my three points (especially the second) are correct.
> >
> > Reviewer ameR

---

### Official Review · Reviewer_HKis · 2021-11-03

**Correctness:** 4
**Technical Novelty And Significance:** 3
**Empirical Novelty And Significance:** 3
**Recommendation:** 6
**Confidence:** 4

**Main Review:**

Strengths:
Paper has a strong motivation for building the dataset. The authors also present a detailed understanding of major applications of their dataset across ML models. It provides a good quantization for the population shift and show with experiments how it impacts domain generalizability. The generalizability of the dataset creation for any dataset with multilabel is another strong point of the paper.

Weakness:
The paper only talks about all advantages for MetaShift that’s derived from Visual Genome data but don’t have any similar comparison or quality analysis for other datasets generated using their 4-step dataset creation, for example, analysis could have been provided on COCO dataset too. The paper also doesn’t address the dependency of dataset performance on metadata, what if there are inconsistencies in the metadata of dataset but images are perfect, how will the dataset creation and performance be impacted for a dataset with such metadata. It would also have been interesting to see the analysis of performance of certain model around underrepresented subsets vs over represented subset, set/characteristics of datum leading to training conflict in general, if there is any pattern in images. Other obvious underlying data bias issue has already been acknowledged in the paper too and I hope the authors will research more into solving it.


**Summary Of The Paper:**

The paper provides a benchmark dataset that can be used for training & evaluation of Machine Learning models. Their contribution is that they have a large collection of 12,868 sets of natural images across 410 classes obtained from visual genome data and its metadata for annotations.  This helps in accounting for large natural shifts in the data. They also provide a way to measure distance between two subsets to quantize the distribution shift between any two of its data sets.
The paper provides a good justification for the need of such a dataset for computer vision tasks and motivate the idea well. It also talks in detail about the steps taken to generate MetaShift from Visual Genome, it also provides a generalization of their 4-step process of dataset creation on any dataset with multilabel, results presented for COCO dataset. The paper further discusses the use of this dataset for two major cases- Evaluating distribution shifts & Assessing training conflicts. They provide the impact of shift distance on the domain generalization by keeping test set same and varying training subsets randomly. Further, it talks about subpopulation shifts where the train and test distribution of same domain with different mixture weights. They show that no algorithm consistently performs better than other algorithms for larger shifts. It provides a detailed understanding of training conflict by analyzing the contribution of each training subset to the change of the validation loss of each validation set during the training process.
Overall, it’s a well written paper about the motivation, use cases, applicability, and generalizability of their proposed data set.


**Summary Of The Review:**

I feel the authors have done a good job highlighting the motivation of such a dataset, steps of creation of the dataset from Visual Genome, paying attention to the generalizability of the approach, discussing the major applications of the dataset in detail. The github link also provides holistic understanding of the work. This dataset will help the research in the field of CV. Once they overcome and address the current weaknesses of the dataset, it will become even better dataset asset.

---

> ### Author Response · Authors · 2021-11-16
> **Response to Reviewer HKis**
>
> Thank you for your thoughtful suggestions and helpful feedback.
>
> **Q: Imperfect or incomplete metadata**
>
> **A:** Even if there is some noise introduced by the imperfect or incomplete metadata, the noise can make the classification tasks more challenging and more realistic, as in subpopulation shifts. For example, it is rare that someone would collect a dog dataset of 100% indoor images, but it is quite possible that the dataset contains 80% indoor images and 20% outdoor images. Therefore, noise from metadata would not invalidate our approach, but make the curated dataset more realistic and challenging. We have included these discussions in Appendix C.3.
>
>
>
> **Q: Analysis on MetaShift built on COCO**
>
> **A:** Following your recommendation, we have added a subpopulation shift experiment on MetaShift constructed from COCO. Similar to Section 4.2, we construct a “Cat vs. Dog” task based on COCO, where the “indoor/outdoor” contexts are spuriously correlated with the class labels. We keep the total size of training data as 3000 images unchanged and only vary the portion of minority groups. We use a balanced test set with 524 images to report both average accuracy and worst group accuracy.
>
> **Table 4: Average accuracy of subpopulation shift on MetaShift from COCO.** p is the percentage of minority groups in the training data.
>
> | Algorithm 	| p=13%     	| p=7%      	| p=1%      	|
> |-----------	|-----------	|-----------	|-----------	|
> | ERM       	| 0.826     	| **0.821** 	| 0.739     	|
> | IRM       	| 0.805     	| 0.794     	| **0.791** 	|
> | GroupDRO  	| 0.821     	| 0.811     	| 0.742     	|
> | CORAL     	| 0.824     	| 0.801     	| 0.750     	|
> | CDANN     	| **0.834** 	| 0.805     	| 0.748     	|
>
>
> **Table 5: Worst group accuracy of subpopulation shift on MetaShift from COCO.** p is the percentage of minority groups in the training data.
>
> | Algorithm 	| p=13%     	| p=7%      	| p=1%      	|
> |-----------	|-----------	|-----------	|-----------	|
> | ERM       	| 0.722     	| **0.698** 	| 0.452     	|
> | IRM       	| 0.765     	| 0.672     	| 0.458     	|
> | GroupDRO  	| 0.730     	| 0.667     	| 0.500     	|
> | CORAL     	| **0.809** 	| 0.695     	| **0.565** 	|
> | CDANN     	| 0.794     	| 0.690     	| 0.527     	|
>
>
> **Results**: Overall, we found that the new results are consistent with our previous findings. When $p$ is large (i.e., the minority group is less underrepresented), ERM outperforms most of the other algorithms in terms of average accuracy. It is also worth noting that no algorithm is a consistent winner compared to other algorithms for large shifts, suggesting that subpopulation shift is an important and challenging task, and that there’s still a lot of room for new methods.
>
> These new experiments also show that MetaShift can flexibly accommodate other source datasets including COCO. We have included these results in Appendix D.
>
>
>
> **Comments on training conflicts:**
> We agree with you that it would be an interesting direction for future work to further visualize the training conflicts for over-represented subsets and under-represented subsets. Furthermore, it would be interesting to vary the percentage of the minority groups (similar to what we did in subpopulation shifts) and see how that would affect the training conflicts.  This is an interesting direction that we will follow up.

---

### Author Response · Authors · 2021-11-16
**General Response**

We thank all the reviewers for your helpful suggestions. We have uploaded a revised paper that addresses your comments. The main updates are:
1. We added new experiments in multi-class classification settings for both (1) domain generation and (2) subpopulation shift.
2. We added new experiments on COCO to demonstrate the utility of MetaShift from COCO.
3. We added t-SNE visualization for the spectral embeddings of a meta-graph.
4. We added additional clarifications on experiment details.
5. We added discussions on imperfect or incomplete meta-data.

---

### Decision · Program_Chairs · 2022-01-20

**Decision:**

Accept (Poster)

**Comment:**

This work studies the impact of distribution shift via a collection of datasets-MetaShift. Reviewers all agreed that this work is simple, effective, and well-motivated, and has key implications, and will be quite useful to the community. There were some concerns about the lack of analysis of MetaShift, and the binary classification setting, which was addressed by the authors’ responses. Thus, I recommend an acceptance.